

# Variations in gut bacterial communities of hooded crane (*Grus monacha*) over spatial-temporal scales

Yuanqiu Dong[1,2], Xingjia Xiang[1,2], Guanghong Zhao[1,2], Yunwei Song[1,2,3] and Lizhi Zhou[1,2]

[1] School of Resources and Environmental Engineering, Anhui University, Hefei, China
[2] Anhui Key Laboratory of Wetland Ecosystem Protection and Restoration, Anhui University, Hefei, China
[3] Anhui Shengjin Lake National Nature Reserve, Chizhou, China

## ABSTRACT

**Background**. Microbes have been recognized as important symbionts to regulate host life. The animal gut harbors abundance and diverse bacteria. Numerous internal and external factors influence intestinal bacterial communities, including diet, seasonal fluctuations and habitat sites. However, the factors that influence the gut bacterial communities of wild bird is poorly characterized.

**Methods**. By high-throughput sequencing and statistical analysis, we investigated the variations in gut bacterial communities of the hooded cranes at three wintering stages in Caizi (CZL) and Shengjin Lake (SJL), which are two shallow lakes in the middle and lower Yangtze River floodplain.

**Results**. Our results revealed significant differences in gut bacterial community structure and diversity among different sampling sites and wintering stages. Seasonal changes have a significant impact on the gut microbe composition of hooded cranes in the two lakes. ANOSIM analysis demonstrated that the samples in CZL had greater differences in the gut bacterial composition than that in SJL. Our data showed strong evidence that the host's gut filtering might be an important factor in shaping bacterial community according to mean nearest taxon distance (MNTD). The PICRUSt analysis showed that the predicted metagenomes associated with the gut microbiome were carbohydrate metabolism, amino acid metabolism and energy metabolism over the entire wintering period at the two lakes.

**Conclusions**. The results demonstrated that both seasonal changes and habitat sites have significant impact on the gut bacterial communities of hooded cranes. In addition, predictive function of gut microbes in hooded cranes varied over time. These results provide new insights into the gut microbial community of the cranes, which serves as a foundation for future studies.

Corresponding author
Lizhi Zhou, zhoulz@ahu.edu.cn

## INTRODUCTION

The gut microbiota confers mutualistic functions involved in substance synthesis and metabolism (*Eberl & Boneca, 2010*), resistance to the intrusion of the pathogen (*Guarner & Malagelada, 2003*; *Koch & Schmid-Hempel, 2011*) and modulation of immune development

(*Eberl & Boneca, 2010*). In vertebrates, the gut harbors diverse and abundant microbes that interact with host and environmental factors to form a complex ecosystem (*Qin et al., 2010*). Previous studies showed that gut microbiota are shaped by diet, life style (*Nicholson et al., 2012*) and environment factors (e.g., seasonal fluctuations and location) (*Hird et al., 2014*).

Birds have unique life history traits that are different from other vertebrates, such as migratory behavior, complex dietary habits, physiological traits and complex network of habitats, all of which may impact gut microbial structure and function (*Kohl, 2012*). Like other vertebrates, the avian gut is colonized with abundant microbes. Previous studies of the intestinal microbiota of birds have mainly focused on ornamental and economical birds, such as parrot (*Waite, Deines & Taylor, 2012*), penguin (*Dewar et al., 2013*) and turkey (*Wilkinson et al., 2017*). However, little information has been reported in wild birds. The rapid development in molecular methodologies has provided new insight for the gut microbiota of birds. Recent research demonstrated that the dominant phyla of the avian gut microbiota were Firmicutes, Proteobacteria, Actinobacteria and Bacteroidetes (*Waite & Taylor, 2014*). Many studies have focused on captive birds which showed that the avian gut microbiota are affected by diet (*Yang, Deng & Cao, 2016*) and gut microbiota exhibited temporal stability (*Kreisinger et al., 2017*). Captivity also has been considered to be an important factor affecting the structure of avian gut microbes. Wild birds remained less studied despite for the association with pathogen transmission. Using the 16s rRNA high through sequencing, we tried to understand the diversity and potential functions of the gut microbiota in hooded cranes and the effects of environmental factors on bird gut microbial communities.

Hooded cranes (*Grus monacha*) are large long-distance migratory colonial wading birds. They are defined as a vulnerable species in the IUCN Red List of Threatened Species (*BirdLife International, 2013*) and Category I key National Protected Wild Animal Species in China. Hooded cranes breed in south-central and south-eastern Siberia and Russia, and winter in Japan, China and South Korea (*Hammerson & Ryan, 2004*). In China, the cranes still inhabit natural lakes and the nearby paddy fields (*Zheng et al., 2015*). The food resources change over spatial–temporal scales. The cranes had to modify their foraging patterns when food recourses change during the wintering periods (*Wan, Zhou & Song, 2016*; *Zheng et al., 2015*). A reduction in the food availability forces animals to modify their foraging patterns (*Zheng et al., 2015*). In the early and mid-winter periods, the most abundant, accessible and frequented food resources were found in paddy fields, while in the late period the more abundant food were available in meadows. Hence, hooded cranes have changed their dietary structure and habitat. In China, hooded cranes are omnivorous birds, but feed mainly on plants (*Huang & Guo, 2015*; *Zhao, Ma & Chen, 2002*).

Shengjin and Caizi Lakes, the two-shallow river-connected lakes in the middle and lower Yangtze River floodplain, respectively, are the most important wintering ground for cranes. These lakes provide the birds with suitable feeding habitats during winter seasons (*Chen et al., 2011*). As a result of the lake degradation in the last decade, the cranes lost many suitable foraging habitats. In this study, we used high-throughput sequencing methods to analyze the gut bacterial community of hooded cranes at three wintering stages in the two lakes. Based on spatial–temporal scales, we tested two hypotheses: (1) Gut bacterial

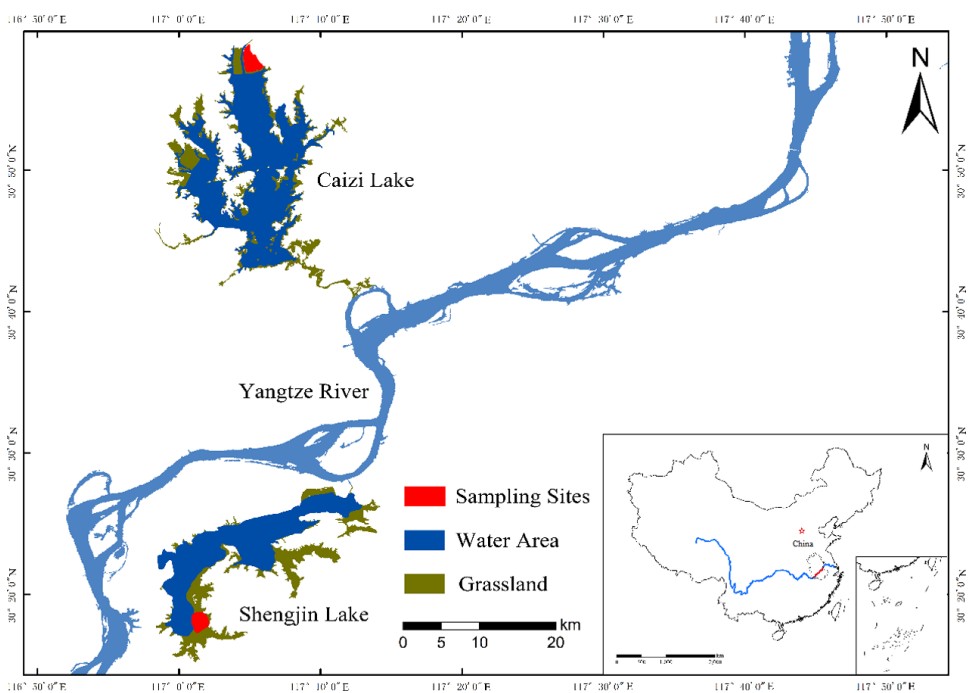

**Figure 1**  **Sampling areas.** The sampling areas of this study.

diversity and composition, as measured by alpha- and beta-diversity, differs significantly in different wintering sites; (2) gut bacterial composition exhibit the same pattern in different wintering location.

# MATERIALS & METHODS

## Ethics statement

Fecal samples were collected after foraging to ensure that the hooded cranes were devoid of human disturbance. It does not involve direct hunting or capture. Permission for the collection of fecal samples was obtained from the local government for wildlife management.

## Study areas

The study was carried out in Shengjin (30.25–30.50°N, 116.92–117.25°E) and Caizi (30.75–30.97°N, 117.00–117.15°E) Lakes, which were located in the middle and lower Yangtze River floodplain (Fig. 1). Both lakes are globally important wintering and stopover habitats for migratory wading birds on the East Asian-Australasian Flyway (*Cao & Fox, 2009*; *Fox et al., 2011a*). Shengjin and Caizi Lakes are river-connected shallow lakes with a northern subtropical monsoon climate. The average annual temperature is 14–18 °C and the average annual rainfall is approximately 1,000–1,400 mm.

The seasonally inundated wetlands provide plenty of food (e.g., *Oryza sativa*, *Polygonum criopolitanu*, and *Potentilla supina*) for the wading birds (*Zhao et al., 2013*; *Zheng et al., 2015*). There are approximately 600 individuals of hooded cranes wintering in the two

lakes each year. The habitat utilization rate of the cranes has shifting over spatial and time scales (*Zhao et al., 2013*; *Zheng et al., 2015*). During the wintering period, shifts in food density and resources triggers cranes to adjust their foraging habitats (*Wan, Zhou & Song, 2016*; *Zhao et al., 2013*; *Zheng et al., 2015*). The wintering period can be divided into three stages according to the migratory rhythm and hydrological processes of the cranes: the early stage from November to December; the middle stage from January to February; and the late stage from March to April (*Zhao et al., 2013*; *Zheng et al., 2015*). The foraging habitats of hooded cranes are relatively stable at certain wintering stage in Shengjin and Caizi Lakes (*Cao & Fox, 2009*).

## Sample collection and preservation

Fecal samples were collected during the three-wintering periods. The foraging sites of the crane flocks were observed with a telescope before sampling to ensure that there were no other species. To avoid human disturbance, the fecal samples were collected immediately upon the completion of foraging. Fecal samples were collected at a minimum distance of approximately 1–2 m to avoid individual sampling repetition (*Zhang & Zhou, 2012*). All samples were obtained from inside the feces to avoid soil contaminants. The samples were kept in a cooler, transported under refrigeration to the lab within several hours and stored at −80 °C. A total of 87 fecal samples were used in this study. Forty-three samples were collected from Shengjin Lake (SJL) and 44 samples from Caizi Lake (CZL). In Shengjin Lake, the samples were 13 in the early wintering period (SJL-E), 14 in the middle wintering period (SJL-M) and 15 in the late wintering period (SJL-L). In Caizi Lake, the samples were 14 in the early wintering period (CZL-E), 15 in the middle wintering period (CZL-M), and 15 in the late wintering period (CZL-L).

## DNA extraction and species determination

DNA was extracted from the fecal samples using the Qiagen QIAamp® DNA Stool Mini Kit following the DNA isolation protocol for pathogens. DNA was eluted in 150 μL of sterilized deionized water and then stored at −80 °C. The COI gene was amplified using the BIRDF1–BIRDR1 primer pair to determine host species. PCR amplification was performed in a 50 μL reaction volume containing 100 ng of fecal DNA, 25 μL of 2× EasyTaq mix (TransGen), 10 μM BIRDF1 (TTCTCCAACCACAAAGACATTGGCAC) and 10 μM BIRDR1 (ACGTGGGAGATAATTCCAAATCCTG). The cycling conditions were as follows: 95 °C for 5 min, followed by 30 cycles of 95 °C for 30 s, 55 °C for 45 s and 72 °C for 90 s, with a final extension period at 72 °C for 10 min. The PCR products were sequenced by Sangon Biotech Co. Ltd. (Shanghai). The sequences were aligned by NCBI. All of the samples were confirmed to contain hooded cranes DNA based on the sequence analysis.

## 16S rRNA sequencing

Primer sets 338F/806R equipped with sequencing adapters and unique identifier tags were used to amplify the V3-V4 hypervariable regions of the bacterial 16S rRNA genes fragments for the Illumina Hiseq platform (PE250). PCR was carried out in 50 μL volume containing 60 ng of fecal DNA, 25 μL of 2× Premix Taq and 1 μL each of the forward and reverse

primers (10 μM). The cycling conditions were as follows: 94 °C for 3 min, followed by 30 cycles of 95 °C for 30 s, 55 °C for 45 s and 72 °C for 45 s, with a final extension at 72x °C for 10 min. Triplicate reaction mixtures per sample were pooled together and purified using the EZNA Gel Extraction Kit (Omega, Norcross, GA, USA). Quantification was performed with a NanoDrop. Sequences were processed using Quantitative Insights Into Microbial Ecology (QIIME). After denoising, poor quality (below an average quality score of 30) and short sequences (shorter than 200 bp) were removed and those sequences which exactly matched with their barcode and primer were kept. The barcode and primer sequences were then removed. Sequences were clustered into OTUs using a 97% identity threshold by UPARSE. Singleton OTUs were deleted and chimers were filtered to remove erroneous OTUs due to sequence errors using the USEARCH in QIIME. The most abundant sequence within each cluster was selected as the representative sequence for that OTU. The taxonomic identify of each OTU was checked against the ribosomal database project (RDP) Classifier. The raw data has been submitted to the SRA database of NCBI with accession number SRP095274.

## Data analysis

Non-metric multidimensional scaling (NMDS) based on the Bray–Curtis dissimilarity (calculated from the relative abundance matrix) and Analyses of Similarities (ANOSIM; permutations = 999) were performed to compare the community composition in different treatments in R 3.4.1 (vegan 2.4-3) (*Dixon, 2010*). In addition, permutational analysis of multivariate dispersions (PERMDISP; permutations = 999) was used to test the gut bacterial community structure heterogeneity varied over spatial–temporal scale. This test was using a Bray–Custis similarity matrix. The significance of PERMDISP was determined via the ANOVA F-statistic to compare among-group differences in the distance from observations to their corresponding group centroids. One-way and two-way ANOVA was used to test observed species, PD whole tree, Shannon and Chao 1 indices among different treatments. The nearest taxon index (NTI) and betaNTI are used to test the assembly processes of the gut bacterial community. NTI was calculated to assess the spatial and temporal changes in bacterial phylogenetic structure using Picante software. The NTI can be used to test for phylogenetic clustering or overdispersion. Positive NTI values and low quantiles ($P < 0.05$) indicate that co-occurring species are more closely related than expected by chance (clustering), whereas positive values and high quantiles ($P > 0.05$) indicate that the co-occurring species are less closely related than expected by chance (overdispersion). Positive betaNTI values indicate greater than expected phylogenetic turnover, and betaNTI was calculated in phylocom. LEfSe (linear discriminant analysis effect size) was used to identifies genomic features characterizing the differences between two or more biological conditions (*Segata et al., 2011*). To detect KEGG pathways with significantly different abundances between the two lakes, LEfSe analysis was used according to the online protocol (http://huttenhower.sph.harvard.edu/galaxy/). Nearest sequenced taxon index (NSTI) was the sum of phylogenetic distances for each organism in the OTU table and the closest genetic relationship of the sequencing reference genome, as measured by the substitutions per site in the 16S rRNA gene and weighted by the frequency of that organism in the OTU table (*Langille et al., 2013*). Functional predictions were made

based on the 16S rRNA OTU membership using PICRUSt (Phylogenetic Investigation of Communities by Reconstruction of Unobserved States) according to the online protocol (http://picrust.github.io/picrust/). One-way ANOVA was used to detect the influence of the sampling site and season on bacterial taxonomy (phylum) variation. All univariate statistical analyses were conducted using SPSS 20.0.

## RESULTS

### Bacterial alpha-diversity

Gut bacterial alpha-diversity was calculated at a depth of 10,000 randomly selected sequences per sample. Alpha-diversity was estimated by the observed species index, phylogenetic diversity, Shannon and Chao 1 index. Two-way ANOVA showed that all alpha diversity was significantly different with time, and only phylogenetic diversity and Chao 1 index were significantly different in spatial variation. Alpha diversity was significantly different under the interaction of wintering period and spatial distance (Table S1). At the early wintering periods, the alpha-diversity of the SJL samples were significantly higher than that of the CZL samples ($P < 0.001$) (Fig. 2), but there was no significant difference in middle and late wintering periods between the two lakes ($P > 0.05$ in both causes). Alpha-diversity was the highest in the middle period across the temporal changes in CZL, and the Chao 1 index of the SJL samples showed significant different across the temporal change (Fg. S1).

### Bacterial community composition

In total, we obtained 4,717,488 reads of high-quality sequences with an average of 46,193 reads was found. Potentially chimeric sequences and singleton OTU (698,694) were then discarded. The dominant gut bacterial phyla were Firmicutes (59.82% ± 32.35%), Proteobacteria (26.82% ± 24.87%), Actinobacteria (5.43% ± 8.19%), Fusobacteria (3.78% ± 12.22%), and Bacteroidetes (2.24% ± 6.21%) (Figs. 3A, 3C). Within Firmicutes, the dominant bacterial classes were *Clostridia* (41.79 ± 31.94%) and *Bacilli* (17.96 ± 21.17%). The classes of *Epsilonproteobacteria* (12.61 ± 21.80%), *Gamaproteobacteri* a (8.69 ± 13.87%) and *Alphaproteobacteria* (4.28 ± 8.04%) were dominant in Proteobacteria (Figs. 3B, 3D). However, the distribution of each taxon among the four groups was uneven, as indicated by Fig. 3. At the family level, the top 30 families were shown for all samples (Fig. S2). At the lower level, only 84.7% of sequences could be assigned to 995 genera. The dominate genera were *Clostridium* (15.5%), *SMB53* (13.62%), *Helicobacter* (11.86%), *Lactobacillus* (7.22%), *Epulopiscium* (4.82%), *Enterococcus* (4.76%), *Fusobacterium* (3.67%), *Pseudomonas* (2.54%), *Turicibacter* (2.08%), *Serratia* (2.03%), *Agrobacterium* (1.77%) and *Lysinibacillus* (1.59%).

One-way ANOVA showed that relative abundance of Firmicutes was significantly higher in SJL samples than in CZL samples, whereas the relative abundance of Proteobacteria from the SJL samples was significantly lower than the CZL samples (Fig. 4, Fg. S3). At Shengjin Lake, the relative abundance of the Firmicutes increased significantly along the winter period. In order to explore the differences in spatial and temporal scale, we conducted LEfSe tests to detect the difference in relative abundance of microbial taxa.

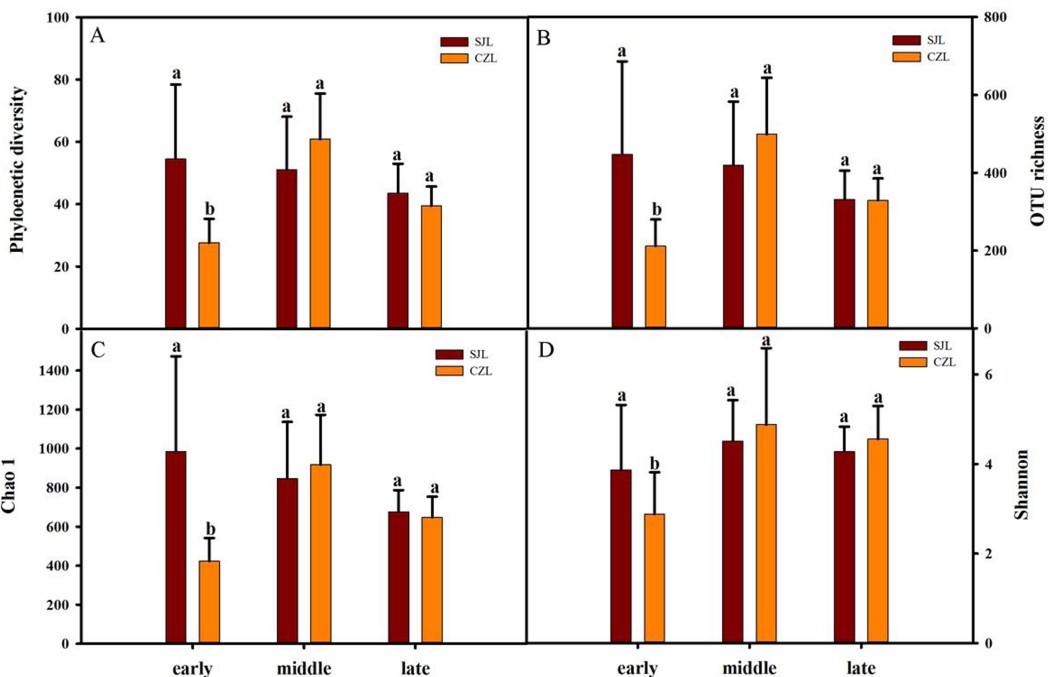

**Figure 2** **Alpha-diversity of different wintering periods.** Variations in alpha-diversity ((A) Phylogenetic diversity, (B) OTU richness, (C) Chao 1 and (D) Shannon) of different wintering periods. Different letters represent significant differences by Tukey's HSD comparisons ($P < 0.05$). Error bars indicate standard deviation.

At Shengjin Lake, four indicator bacterial taxa (i.e., *Pseudomonadaceae, Pseudomonas, Sphingobacteriales, Sphingobacterii*) were found in the gut of the hooded crane in SJL-E samples and five indicator bacterial taxa (i.e., *Clostridiales, Clostridia, Lachnospiraceae, Epulopiscium, Peptostreptococcaceae*) were found in the gut of hooded crane in SJL-L samples (Figs. 5A, 5B). At Caizi Lake, three indicator bacterial taxa (*Fusobacteriaceae, Fusobacterium, Peptostreptococcaceae*) were found in the CZL-E samples, four indicator bacterial taxa (i.e., *Sphingobacteriaceae, Sphingobacterii, Sphingobacteriales, Microccaceae*) were found in the CZL-M samples and 11 indicator bacterial taxa (*Gammaproteobacteria, Enterobacteriaceae, Enterobacteriales, Bacilli, Lactobacillales, Enterococcaceae, Enterococcus, Lactobacillales, Enterococcus, Lactobacillus, Lactobacillaceae, Enterobacter*) were found in the CZL-L samples (Figs. 5C, 5D).

The bacterial community compositions were significantly different between the guts of hooded crane from SJL samples and CZL samples. NMDS revealed that SJL samples tended to be less different compared to the CZL samples (Fig. 6E). ANOSIM analysis confirmed that seasonal changes had a significant impact on the microbial composition of the hooded cranes in two lakes (Table 1). The bacterial community composition of SJL samples and CZL samples were significantly different ($P = 0.001$) in the three wintering periods (Table 1), the bacterial community compositions in different wintering period were significantly different in both Shengjin and Caizi Lake ($R = 0.238, P = 0.001$) (Table 1).

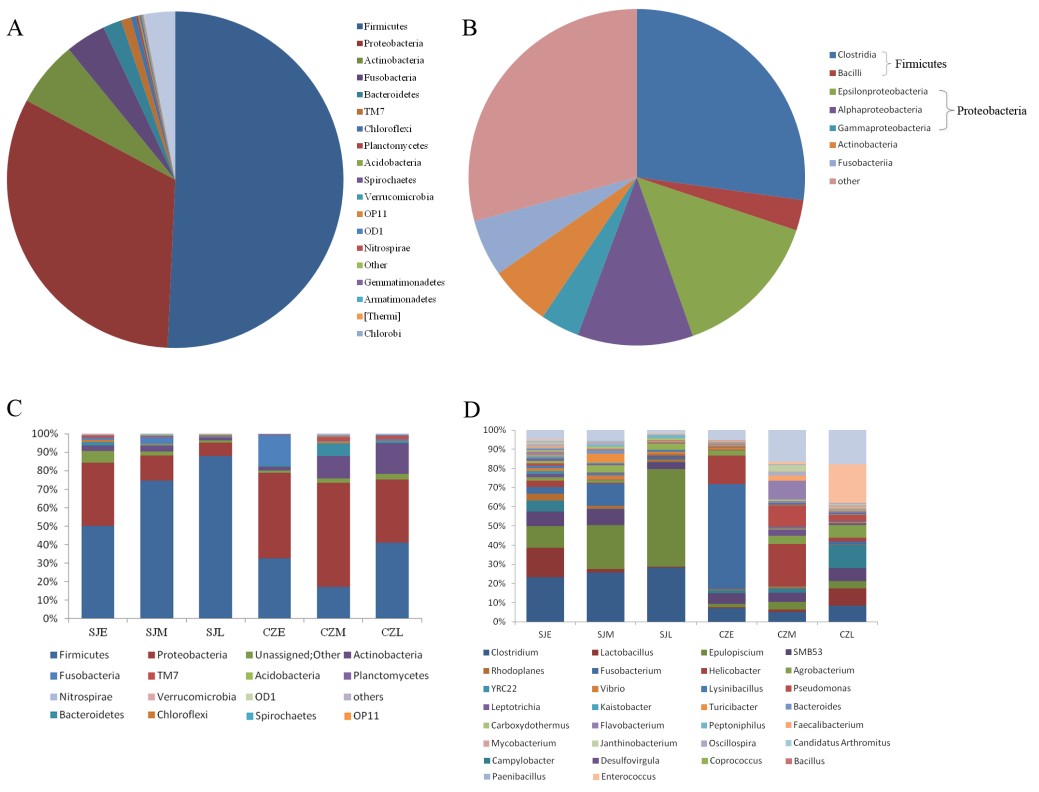

**Figure 3 Relative abundance of gut bacteria at the phylum, class and genus levels at two lakes.** (A) Relative abundance of the dominant phyla in all samples. (B) Relative abundance of the dominant classes in all samples; Rings represent corresponding phylum (Firmicutes in blue and Proteobacteria in red) for each of the most frequently represented class. (C) Relative abundance at the phylum and (D) relative abundance at the genera at different periods in the two lakes.

According to PERMDISP, average distance to the corresponding group centroid have no significant differences among different wintering periods at Shengjin Lake ($P = 0.167$). However, mean distance to the corresponding group centroid have significant differences among different wintering periods at Caizi Lake ($P < 0.001$). Furthermore, no significant differences could be detected among average distance to the corresponding group centroid of gut bacterial communities in different sample collected sites.

## Assemblage processes of the gut bacterial community

NTI was used to evaluate the gut bacterial phylogenetic structure. All of the NTI values were positive, which showed that the bacterial communities were phylogenetically clustered (Fig. 7, Fig. S4). At the early wintering stage, the CZL-E samples NTI values were less positive compared to the SJL-E samples, which indicated that phylogenetic clustering was weakest in the CZL-E samples (Fig. 7). In addition, the NTI values of SJL-L samples were less positive compared to the CZL-L samples at late wintering stage, which indicated that phylogenetic clustering was weakest in the SJL-L samples. Phylogenetic clustering was similar in SJL samples during three wintering periods, whereas CZL-L samples phylogenetic clustering were more similar in CZL samples.

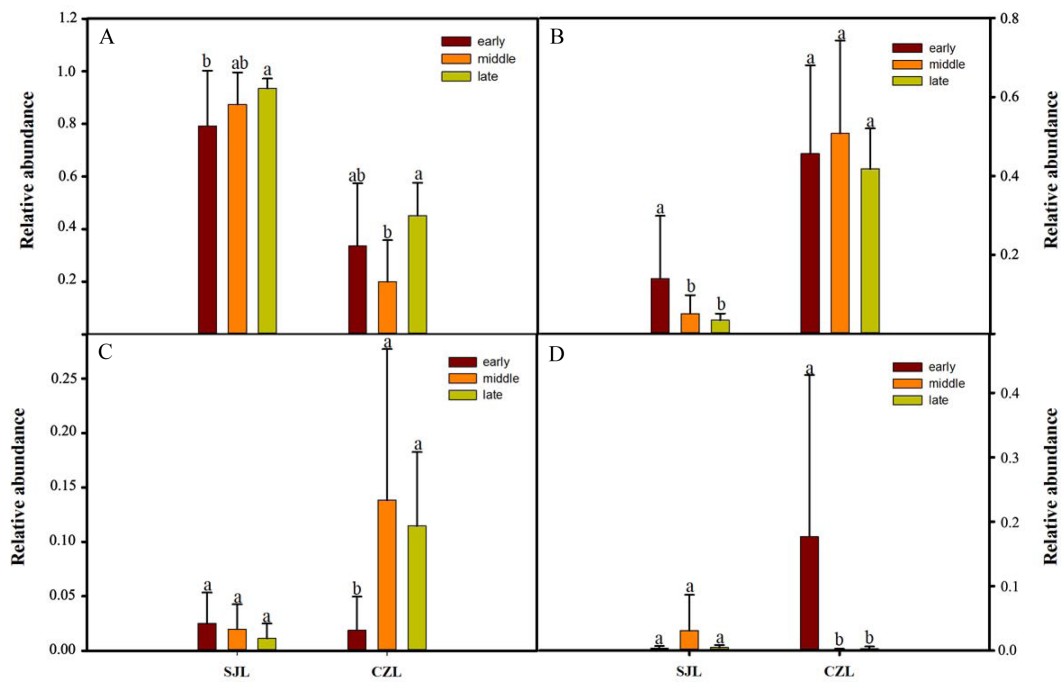

**Figure 4 Relative abundance of the dominant phyla (A) Firmicutes, (B) Ptoteobacteria, (C) Actinobacteria, (D) Fusobacteria) in different wintering periods.** Samples are grouped according to sampling location and wintering periods. Different letters represent significant differences by Tukey's HSD comparisons ($P < 0.05$). Error bars indicate standard deviation.

## Variation in predicted metagenomes between the two lakes

In addition, NSTI also influenced PICRUSt accuracy. The NSTI for our samples was $0.18 \pm 0.07$ similar to the previously reported analyses in soils (mean NSTI = $0.17 \pm 0.02$). In this study, a total of 41 functional genes were predicted in hooded crane population. The majority of functions were membrane transport (12.63%), carbohydrate metabolism (9.36%), amino acid metabolism (9.19%), replication and repair (8.08%), energy metabolism (7.10%) and translation (5.18%) during the wintering period. ANOSIM analysis revealed that the potential functions of the gut bacterial communities of the two lakes were significantly different during early and late wintering periods ($P = 0.001$) (Table 2). However, during the middle wintering period, there was little difference in the potential functions of the gut bacterial in the two lakes ($P = 0.116$). In early wintering period, hooded crane from Caizi Lake were predicted to have a microbiota with enhanced capability for metabolic disease, carbohydrate metabolism, amino acid metabolism, circulatory transport, cellular processes and singling and a lower rate of microbial genes associated with genetic information processing (Table S2). In middle wintering period, hooded crane from Caizi Lake were predicted to have a microbiota with enhanced capability for amino acid metabolism, sensory system, transport and catabolism, membrane transport and genetic information processing. When comparing predicted metagenomes in the late period, the microbiota of hooded crane from Caizi Lake was predicted to have a capability

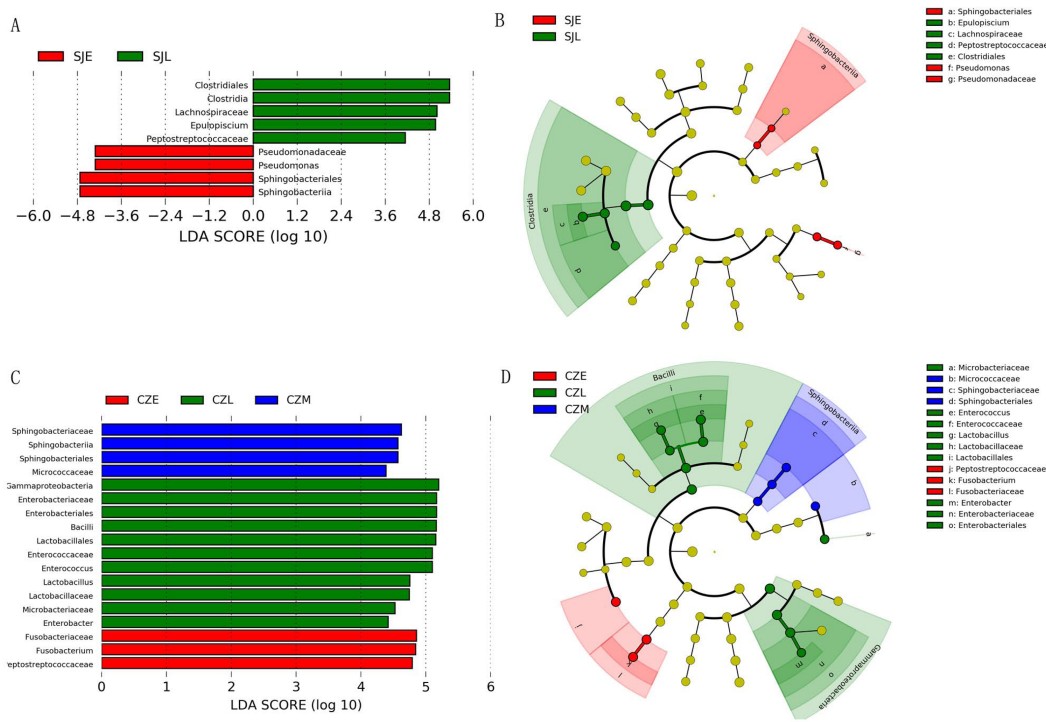

**Figure 5** **LEFSe analysis of the hooded Crane guts bacteria in different wintering periods (LDA > 2, P < 0.05).** (A) LDA score representing the taxonomic with significant difference in different wintering periods at the Shengjin Lake. (B) Cladogram representing the taxonomic hierarchical structure of the phylotype biomakera identified among three wintering periods from the Shengjin Lake; Red, phylotypes statistically overrepresented in early wintering period; Green, phylotypes statistically overrepresented in late wintering period. (C) LDA score representing the taxonomic with significant difference in different wintering periods at the Caizi Lake. (D) Cladogram represting the taxonomic hierarchical structure of the phylotype biomakera identified among three wintering periods from the Caizi Lake; Red, phylotypes statistically overrepresented in early wintering period; Green, phylotypes statistically overrepresented in late wintering period; Blue, phylotypes statistically overrepresented in middle wintering period.

for metabolic disease, carbohydrate metabolism, amino acid metabolism, replication and repair and glycan biosynthesis and metabolism.

## DISCUSSION

Hooded cranes have to modify their foraging behavior to adapt to variations in food availability over a spatial–temporal scale as a result of wetland loss and degradation (*Wan, Zhou & Song, 2016*; *Zheng et al., 2015*). Having a diverse gut microbiome may be another adaptive mechanism. In this study, the gut bacterial composition and structure of the wintering hooded Crane from the Shengjin and Caizi Lakes were explored. Differences in microbial community structures and interactions were also identified.

The gut microbial community of the hooded crane was dominated by Proteobacteria, Firmicutes and Actinobacteria, which was similar to that of mammals (*Waite & Taylor, 2014*). The phylum of Firmicutes was dominated in hooded crane gut at the two lakes and was consistent with the many avian species (*Dewar et al., 2014*; *Waite & Taylor, 2014*;

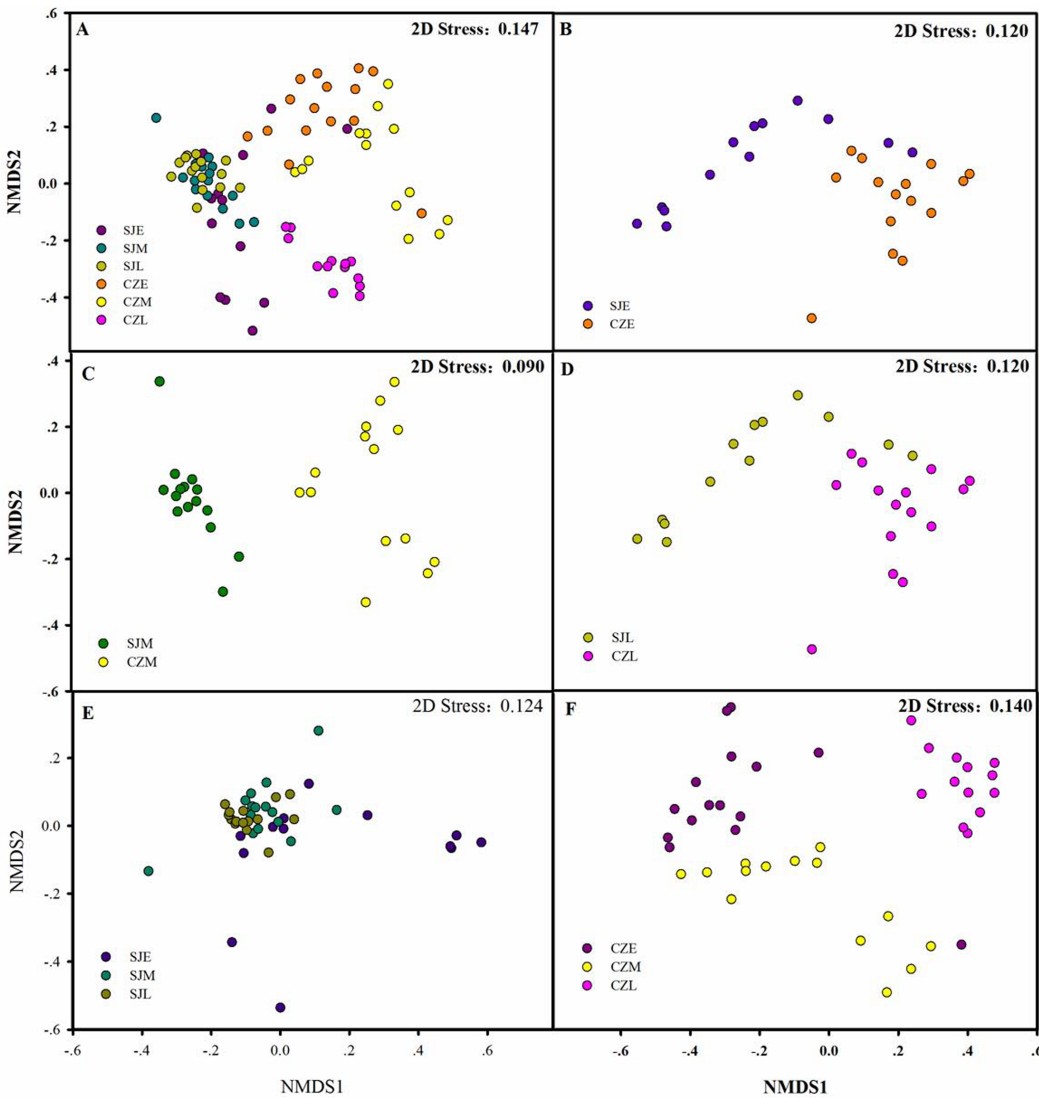

**Figure 6** **Differences in gut bacterial community composition.** (A) Non-metric multidimensional scaling plot bacterial community composition from Shengjin and Caizi Lakes at three wintering periods. (B) Non-metric multidimensional scaling plot showing bacterial community composition from Shengjin and Caizi Lakes at the early wintering period. (C) Non-metric multidimensional scaling plot for the hooded crane from Shengjin and Caizi Lakes at the middle wintering period. (D) Non-metric multidimensional scaling plot for the hooded crane from Shengjin and Caizi Lakes at the late wintering period. (E) Non-metric multidimensional scaling plot for the hooded crane from Shengjin Lake at three wintering periods. (F) Non-metric multidimensional scaling plot for the hooded crane from Caizi Lake at three wintering periods.

*Wilkinson et al., 2017*). Firmicutes are associated with the breakdown of carbohydrates, polysaccharides, sugars and fatty acids, which are utilized by the host as energy sources (*Flint et al., 2008*). The *clostridium* genus belonging to Firmicutes can digest simple carbohydrates (*Aristilde, 2017*; *Bäckhed et al., 2004*) as well as complex polysaccharides (*Aristilde, 2017*; *Ramos et al., 2015*), which may lead to high proportion of carbohydrates metabolism.

**Table 1 ANOSIM analysis.** Differences in the microbial community composition based on the similarity test of ANOSIM.

| | Temporal variation | | | Spatial variation | |
|---|---|---|---|---|---|
| | R | P | | R | P |
| SJE vs. SJM | 0.264 | 0.001 | SJE vs CZE | 0.591 | 0.001 |
| SJM vs. SJL | 0.124 | 0.009 | SJM vs CZM | 0.85 | 0.001 |
| CZE vs. CZM | 0.4 | 0.001 | SJL vs CZL | 0.967 | 0.001 |
| CZM vs. CZL | 0.845 | 0.001 | | | |
| SJE vs. SJM vs. SJL | 0.238 | 0.001 | | | |
| CZE vs. CZM vs. CZL | 0.739 | 0.001 | | | |

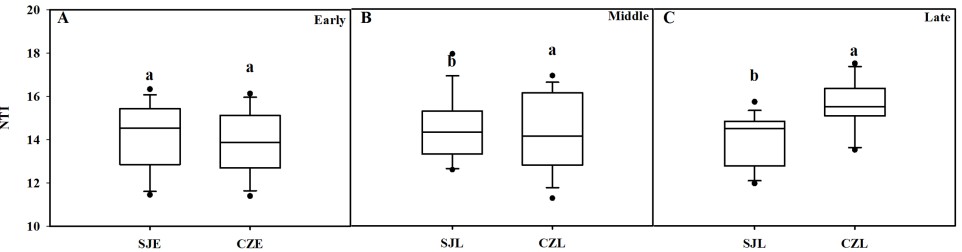

**Figure 7 Bacterial phylogenetic structure evaluated by the NTI in different wintering periods.** (A) At the early wintering period, (B) at the middle wintering period, (C) at the late wintering period.

All above results indicated that the hooded cranes, like other waterbirds, feed on high-energy foods such as *Vallisneria* tubers (*Fox et al., 2011a*; *Xu et al., 2008*). In order to survive in the dry and cold weather, the hooded cranes need considerable amounts of energy (*Cai, Huettmann & Guo, 2014*; *Fox et al., 2011a*). However, Proteobacteria plays an important role in energy accumulation (*Bryant & Small, 1956*; *Chevalier et al., 2015*), and the high proportion of Proteobacteria may be due to the cranes need a lot of energy to cope with the cold winter. The classes of *Epsilonproteobacteria*, *Gamaproteobacteri* a and *Alphaproteobacteria* were dominant in Proteobacteria, similar to other avian, such as kittiwake, bean goose, kakapo (*Van Dongen et al., 2013*; *Waite, Deines & Taylor, 2012*; *Yang, Deng & Cao, 2016*). The dominant families were *Xanthomonadaceae* and *Comamonadaceae*. These two families are strong competitors with flexible metabolism and are considered essential for host's digestion in poor nutritional diet. The phylum Fusobacteria associated with butyrate production, with butyrate playing an important role in ion absorption and immune regulation. It is also an important anti-inflammatory agent (*Canani et al., 2011*). The phylum of Actinobacteria also found in the hooded cranes and associated with pathogens, such as the genera of *Nocardia* and *Rhodococcus* (*Santos et al., 2012*). The habitats of domestic waterbirds overlapping with hooded cranes at the two lakes, and large assemblance of migratory birds may represent a source of pathogenic microbes that can be transmitted by each other through feces (*Zhao, Ma & Chen, 2002*).

However, the detailed composition of these phyla was notably altered with space–time according to our study. The samples from Shengjin Lake were dominated by Firmicutes,

**Table 2  ANOSIM analysis.** Differences in the microbial functions based on the similarity test of ANOSIM.

| | Temporal variation | | | Spatial variation | |
|---|---|---|---|---|---|
| | **R** | **P** | | **R** | **P** |
| SJE vs SJM vs SJL | 0.181 | 0.005 | SJE vs CZE | 0.444 | 0.001 |
| CZE vs CZM vs CZL | 0.218 | 0.001 | SJM vs CZM | 0.058 | 0.116 |
| | | | SJL vs CZL | 0.221 | 0.001 |

while those from Caizi Lake were dominated by Firmicutes and Proteobacteria. Compared to Shengjin Lake, the samples from the Caizi Lake appear to be enriched in Proteobacteria and the relative abundance of Firmicutes decreased (Fig. S1). Increased the relative abundance of Proteobacteria in the gut microbiota increase the risk of inflammatory bowel disease (*Zhou et al., 2016*). According to LEfse analysis, genus of *Pseudomonas* was enriched in the gut of hooded crane from SJL-E samples. Pervious study shows that the *Pseudomonas* genus codes for levansucrases (*Visnapuu, Mardo & Alamäe, 2015*). We also found the class of sphingobacteriales were highest in the gut of hooded crane from SJL-E samples and CZL-M samples (Fig. 5) because of the increase in polysaccharide (*Carney et al., 2014*). In addition, there was no significant difference in the alpha diversity of the SJL samples in different wintering periods, while the alpha diversity of the CZL was significantly difference in samples in different wintering periods (Fig. S1). Furthermore, significant differences in gut microbial communities were identified, as reflected by NMDS clustering and microbial interactions (Fig. 4, Table 1). These results showed that the seasonal and temporal change may be important factors in shaping host bacterial structure. In recent years, grazing animal waste and poultry litter effects on the environment (*Sauer et al., 1999*). As most members of genera (i.e., *Agrobacterium*) are present in soil, these bacteria may have originated from the environment, implying horizontal transmission could also influence the structure of animal gut microbiota (*Yang, Deng & Cao, 2016*). Thus, grazing animal waste and poultry litter also affects the structure of the gut microbiota. Seasonal fluctuations also had a significant effect on the composition of gut microbes at the two lakes.

In addition, all the bacterial community assemblages showed significant phylogenic clustering, indicating that bacterial communities were strongly structured by gut environment filtering (*Yan et al., 2016*). Although the bacterial community compositions showed significantly different between the two sites in the early wintering period, however, the spatial changes did not affect bacterial phylogenetic clustering, while gut environmental filtering influenced bacterial communities in both lakes in the middle and late periods. The effect of temporal changes on bacterial phylogenetic structure was not consistent in Shengjin and Caizi lakes. These results suggest that the environment factors can influence the composition of bacterial community and might be an important factor for bacterial phylogenetic structure.

The bacterial community in the guts of hooded cranes may have many important functions. In this study, PICRUSt were used to deduce potential gene profiles from 16S rRNA sequencing. The result showed that the most abundant functional classes were related to membrane transport, carbohydrate, amino acid and energy metabolism. The higher

proportion of energy metabolism pathway-related predicted genes may meet the hooded crane's energy needs for flight. The energy metabolism pathways were much higher in the late wintering than in the other two periods, which may be induced by the fact that the food density gradually decreases over winter period, leading to an increase in the foraging efforts of hooded cranes (*Wan, Zhou & Song, 2016*). Moreover, the PICRUSt results revealed that the glycan biosynthesis and metabolism-related genes were present in both lakes but with higher proportion at Caizi Lake. Therefore, we propose that this pathway in the gut microbiota may significantly contribute to increase digestive efficiency and assimilation, which may play an important role in providing energy and nutrients to cope with the cold weather (*Wan, Zhou & Song, 2016*).

## CONCLUSIONS

In this study, we showed that gut bacterial community composition and diversity of hooded cranes changed over spatial–temporal scales. We found that spatial–temporal changes might be important factors to influence gut bacterial community composition. In addition, dietary seasonal fluctuation also affected the gut bacterial community composition. This study provides a foundation understanding of gut bacterial community composition and potentially bacterial functions in hooded cranes. Future work should focus on how these actual functions relate to the gut microbial community composition.

## ACKNOWLEDGEMENTS

We appreciate the help from the staff of Shengjin Lake Nature Reserve Bureau and local villagers. We are grateful to Binghua Sun, Nazia Mehtab and Yang Li for their assistance in data analysis.

### Funding

This work was supported by funding from the National Natural Science Foundation of China (Grant No. 31772485 and 31472020) and the Graduate Student Innovation Research Projects of Anhui University (YQH100237). The funders had no role in study design, data collection and analysis, decision to publish, or preparation of the manuscript.

### Grant Disclosures

The following grant information was disclosed by the authors:
National Natural Science Foundation of China: Grant No. 31772485 and 31472020.
Graduate Student Innovation Research Projects of Anhui University (YQH100237).

### Competing Interests

The authors declare there are no competing interests.

## Author Contributions

- Yuanqiu Dong conceived and designed the experiments, performed the experiments, analyzed the data, contributed reagents/materials/analysis tools, prepared figures and/or tables, authored or reviewed drafts of the paper, approved the final draft.
- Xingjia Xiang conceived and designed the experiments, analyzed the data, contributed analysis tools, prepared figures and/or tables, authored or reviewed drafts of the paper, approved the final draft.
- Guanghong Zhao and Yunwei Song the sample collection.
- Lizhi Zhou conceived and designed the experiments, performed the experiments, contributed reagents/materials/analysis tools, authored or reviewed drafts of the paper, approved the final draft.

## Data Availability

NCBI SRA accession number: SRP095274.

## Supplemental Information

Supplemental information for this article can be found online at http://dx.doi.org/10.7717/peerj.7045#supplemental-information.

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
