# Peer review of "Variations in gut bacterial communities of hooded crane (Grus monacha) over spatial-temporal scales"

_PeerJ, doi:10.7717/peerj.7045_

## Round 0.1 · original submission · Major Revisions

Please provide a point-by-point response to each of the three reviewers' comments, including those provided in the annotated manuscript provided by reviewer 1.

·

Basic reporting

The article provides an original investigation into the seasonal and geographical effects on the gut microbiome of wintering Hooded cranes.
The literature and background are sufficient.

There are some grammar and structural issues with the wording of the article (please see comments), but overall the article is well structured.

Experimental design

The originality of the research fits within the aims and scope of the journal and the research is well defined and meaningful and highlights an important knowledge gap.
The experimental design and statistical analysis is appropriate for this study.

Validity of the findings

The conclusions of this study are well stated and linked to previous research and the findings valid. There are some areas that need addressing (please see comments).

Additional comments

Please address the comments highlighted in the pdf copy of your manuscript.

·

Basic reporting

1) Detailed editing of the English language and grammar is required throughout the text. Most notably, ensure correct usage of singular and plural forms of words, apostrophes for possession, and pronouns (e.g., within the Introduction, see lines 46, 56, 59-61, 65, 76, 80). One specific example is the inconsistency with which microbiota is used as a singular or plural word.

2) For Figure 3, a heat map with the profile data for all samples (all the individual samples from the 2 sites across the 3 wintering phases) at the family/genus level would be a valuable addition to the manuscript.

3) The LEfSe results are presented in lines 237-241. However, this presentation is minimal. It would be beneficial to further explain the indicator taxa. Of note, the indicator taxa between early and late wintering stages are entirely different for the two sampling locations. This should be explicitly addressed here or in the Discussion, as it does not support the idea that hooded cranes are experiencing consistent shifts in their microbiomes across the winter.

4) In the last paragraph of the Results section, in lines 270-275, it is noted that predicted functional gene profiles of fecal samples differed between sampling sites and wintering phases. However, the indicator genes/pathways/functions for the different sample groups are not currently explained or presented at all. This should be addressed.

5) The Discussion section is currently very general, largely because the focus is on phylum-level variation in fecal bacterial profiles. The Discussion would be more informative if the focus was on more fine-scale taxonomic differences among sample groups (i.e. OTU/genus/family level).

6) It is noted in the Introduction that food density and resources change across the wintering period for these cranes (lines 116-188). This should be further explained here.

Experimental design

1) The research questions are well defined: Assess if the gut bacterial diversity of hooded cranes 1) differs between two wintering sites; 2) is consistently influenced by seasonal changes over winter; 3) responds to dietary changes. However, the third research question was not specifically addressed in this study because data on dietary resources and consumption were not collected. Therefore, the study actually had two research questions. Inferences are here drawn about the originally proposed 3rd question from the results of the 2nd question.

2) In lines 163-165, it is noted that sequences were processed using a coupled protocol of both mothur and QIIME programs. However, it is not currently clear what this entails, which processing steps were completed with which program, or why this approach was taken.

Validity of the findings

1) For univariate (ANOVA) and multivariate (ANOSIM) analyses, it is not clear why the authors are not using two-way models (site X wintering phase). Two-way analyses can be run in the same packages the authors have used to conduct the numerous one-way analyses.

2) In line 243, it is observed that there was greater dispersion in the fecal bacterial profiles during the late wintering period at both sampling sites. However, this is not clearly evident in Figure 6. This statement should be statistically evaluated using permutation of dispersions analysis (also available in the vegan package in R).

Additional comments

Include a specific estimate of the time that fecal samples were in the transport cooler between collection and storage in the -80C freezer (see lines 131-133). “As soon as possible” is too vague for the purposes of replication and/or interpretation.

In line 180, it indicates that alpha diversity was characterized using the Simpson Index, however, Simpson data are not included in the manuscript. Either add the data or remove Simpson from the list of diversity measures provided here.

In the legend for Figure 5, panels C and D are not explained and B is currently incorrectly explained (actually C).

In Figure 6, panels A-F are not currently labeled.

Reviewer 3 ·

Basic reporting

This study used high-throughput sequencing of the 16S rRNA gene to analyze the gut microbial communities of Hooded cranes at three wintering stages in the two lakes in China. The study is largely descriptive. I have several methodological questions about the work, which I describe below. In addition, the manuscript contains several grammatical problems, and it should be copy edited by a native speaker of English. It is not clear whether the data have been made publicly available.

Experimental design

Methodological issues in order of appearance in the manuscript

Figure 1 - I am not sure where the study site is located in China. Can you add detail to the legend or the figure itself to explain where the larger map is located in the inset map of China? What is the imporance of the smallest inset map, which appears to be of a chain of islands?

Was a bead beating step included in DNA extraction? Bead beating is typically included in microbiome studies to increase detection of bacteria with cell walls. If a bead beating step was not included, please explain the implications for interpeting your results.

In your pipeline, please explain how many total reads you started with in the raw sequencing data and how many reads were lost during quality filtering steps.

Please describe how you evaluated the quality of your PICRUSt analyses. These analyses rely on several assumptions which may not be sound for your system. See more at:
https://picrust.github.io/picrust/tutorials/quality_control.html

In your analyses of alpha diversity and Figure 2, please explain how you controlled for read count. More diversity is expected with higher read counts; did you control for read count by adding it as a factor in your models or rarefying the data?

Please create a version of Figure 3 that shows how taxonomic composition varies between samples. In the current version, it's not clear how the abundance of phyla and classes varied between different samples. Please revise this figure to make it a stacked bar chart with one bar for each sample in your data set.

Validity of the findings

In the introduction, you specify three hypotheses. Your third hypothesis states that, "Dietary changes in different seasonal affect the gut bacteria of Hooded cranes." However, you have no data on dietary differences between birds and so you cannot test this hypothesis. Please remove this hypothesis from the manuscript.

Moreover, while you should be able to test hypothesis 2 ("Gut bacterial are influenced by seasonal changes and exhibit the same pattern in different wintering location") you do not clearly test any direct predictions of this hypothesis in the results or discussion or state whether this hypothesis was supported or rejected by your data.

The discussion fails to place the work in the broader context of what is known about avian gut microbiota. Please expand your review of the literature in the discussion to compare your results to what has been observed in other wild birds.

Additional comments

Minor comments

Abstract line 21 - "bird" should be "birds"

Abstract line 28 - "ANOSIM analysis explored that" should be "ANOSIM analysis revealed that"

Abstract line 30 - "to shape" should be "in shaping"

Abstract line 31 - The following sentence is missing a word: "In addition, PICRUSt analysis revealed that the predicted metagenomes associated with carbohydrate metabolism, amino acid metabolism and energy metabolism over the entire wintering period at the two lakes"

Introduction line 56 - "birds gut" should be "bird guts" or "the gastrointestinal tract of birds"

Introduction line 59 - this sentence has grammatical problems, "Past decade with rapid development in molecular methodologies, which has been provided new insight for the gut microbial of birds."

The grammatical/language issues in this paper were so numerous that I stopped recording them. This paper will need to be copy editied by a native speaker of English.

Line 100 - did you receive a permit number with the permission to collect samples, or was a permit not issued or unecessary?

Line 190 - Please spell out the full acronym for LEfSe (linear discriminant analysis effect size)

Line 218 - you are missing a comma in "4018,794". Should this number be 4,018,794?

---

## Round 0.2 · Major Revisions

Please re-check your response to the reviewer comments against the revised version of your manuscript. I am having difficulty matching the line numbers in your response to reviewers with the line numbers in either the marked or clean version of your revised manuscript. This makes it very difficult for me to see whether you have sufficiently addressed the reviewers' comments.

---

## Round 0.3 · accepted · Accept

Thank you for addressing the detailed comments of the three reviewers, and I believe the revised manuscript is much improved.